# Prognostic Significance of Sarcopenia in Patients with Unresectable Advanced Esophageal Cancer

**DOI:** 10.3390/jcm8101647

**Published:** 2019-10-09

**Authors:** Sachiyo Onishi, Masahiro Tajika, Tsutomu Tanaka, Yutaka Hirayama, Kazuo Hara, Nobumasa Mizuno, Takamichi Kuwahara, Nozomi Okuno, Yoshitaka Inaba, Takeshi Kodaira, Tetsuya Abe, Kei Muro, Masahito Shimizu, Yasumasa Niwa

**Affiliations:** 1Department of Endoscopy, Aichi Cancer Center Hospital, 1-1 Kanokoden, Chikusa-ku, Nagoya 464-8681, Japan; soonishi@aichi-cc.jp (S.O.); tstanaka@aichi-cc.jp (T.T.); yhirayama@aichi-cc.jp (Y.H.); yniwa@aichi-cc.jp (Y.N.); 2Department of Gastroenterology, Aichi Cancer Center Hospital, 1-1 Kanokoden, Chikusa-ku, Nagoya 464-8681, Japan; khara@aichi-cc.jp (K.H.); nobumasa@aichi-cc.jp (N.M.); kuwa_tak@aichi-cc.jp (T.K.); nokuno@aichi-cc.jp (N.O.); 3Department of Diagnostic and Interventional Radiology, Aichi Cancer Center Hospital, 1-1 Kanokoden, Chikusa-ku, Nagoya 464-8681, Japan; 105824@aichi-cc.jp; 4Department of Radiation Oncology, Aichi Cancer Center Hospital, 1-1 Kanokoden, Chikusa-ku, Nagoya 464-8681, Japan; 109103@aichi-cc.jp; 5Department of Gastroenterological Surgery, Aichi Cancer Center Hospital, 1-1 Kanokoden, Chikusa-ku, Nagoya 464-8681, Japan; tabe@aichi-cc.jp; 6Department of Clinical Oncology, Aichi Cancer Center Hospital, 1-1 Kanokoden, Chikusa-ku, Nagoya 464-8681, Japan; kmuro@aichi-cc.jp; 7Department of Gastroenterology/Internal Medicine, Gifu University Graduate School of Medicine, 1-1 Yanagido, Gifu 501-1194, Japan; shimim-gif@umin.ac.jp

**Keywords:** sarcopenia, esophageal cancer, skeletal mass index, malnutrition, aspiration pneumonia, prognosis

## Abstract

The prognostic significance of sarcopenia in unresectable advanced esophageal cancer remains unclear. Our study retrospectively evaluated 176 consecutive Japanese patients with esophageal squamous cell carcinoma who had been diagnosed with unresectable advanced cancer in Aichi Cancer Center Hospital between January 2007 and December 2014. Skeletal muscle mass was calculated from abdominal computed tomography (CT) scans before treatment, and patients were divided into sarcopenic and non-sarcopenic groups. Sarcopenia was present in 101 patients (57.4%). Eighty-two patients in the sarcopenic group and 63 patients in the non-sarcopenic group died during follow-up (mean: 20.3 months). The overall survival (OS) rate was significantly lower in the sarcopenic group compared to the non-sarcopenic group (2-year OS: 9.8% vs. 23.7%, *p* < 0.01). Cox regression analysis revealed only pretreatment sarcopenia as an independent prognostic factor (hazard ratio (HR): 1.48, 95% confidence interval (CI): 1.04–2.10, *p* = 0.03). In the sarcopenic group, withdrawn cases, for whom the planned treatment was discontinued for some reason, showed a significantly lower OS rate compared to complete cases (1-year OS: 11.0% vs. 59.9%, *p* < 0.01). The most common reason for discontinuation was aspiration pneumonia (64.5%). Presence of sarcopenia was an independent prognostic factor for unresectable advanced esophageal cancer. Identifying the presence of sarcopenia prior to treatment may improve the prognosis.

## 1. Introduction

Esophageal cancer is the sixth most common cause of cancer-related deaths worldwide [1]. Although curative surgery is the standard treatment for patients with resectable esophageal cancer, about 50% of patients with esophageal cancer have unresectable advanced tumor due to radiographically visible metastases. Treatment for such patients is commonly palliative chemotherapy to prolong survival, but response rates are 35–45%, and survival is typically prolonged by only a few months [2]. The 5-year overall survival (OS) rate is extremely low, at 15–25% [3]. Because of obstructive symptoms related to unresectable disease before treatment, patients often cannot eat sufficiently, and the nutritional condition thus deteriorates.

Sarcopenia is characterized by loss of skeletal muscle mass, skeletal muscle strength, and physical performance. Sarcopenia has been shown to impair survival in geriatric noncancer populations [4]. Sarcopenia is classified into primary sarcopenia, which is caused by aging, and secondary sarcopenia, which includes activity-related, disease-related, and nutrition-related sarcopenia [4,5]. Secondary sarcopenia related to malignant disease has been reported in recent years and has been recognized as indicating poor prognosis in patients with various malignancies [6,7,8,9,10,11,12], including lung, gastrointestinal, pancreatic, hepatic, breast, urothelial, and colorectal cancers. In esophageal cancer, the prognostic value of sarcopenia in surgically treated patients has not been clearly established, despite investigation in numerous cohort studies [13,14,15,16,17,18,19]. However, a recent comprehensive systematic review and meta-analysis of 11 cohort studies consisting of a total of 1520 patients with esophageal cancer after esophagectomy showed that patients with sarcopenia correlated significantly with lower 3- and 5-year OS rates compared to patients without sarcopenia [20,21]. Moreover, preoperative sarcopenia was identified as an independent predictor of poor OS and disease-free survival (DFS) in patients with surgically treated esophageal cancer, regardless of whether patients received preoperative treatment or how sarcopenia was defined. On the other hand, little is known about the relationship between sarcopenia and prognosis in patients with unresectable advanced esophageal cancer. We therefore examined the prevalence of sarcopenia in unresectable advanced esophageal cancer and investigated the impact on prognosis.

## 2. Materials and Methods

### 2.1. Study Population

We investigated 176 consecutive inpatients with unresectable advanced esophageal squamous cell carcinoma treated at Aichi Cancer Center Hospital between April 2007 and December 2014. All patients were investigated on their first hospitalization to receive initial treatment and underwent computed tomography (CT) of the whole body from neck to pelvis for diagnostic purposes. The study protocol was approved by the ethics committee at Aichi Cancer Center Hospital (2017-1-360 observation study) and performed in accordance with the 1975 Declaration of Helsinki as revised in 1983.

### 2.2. Treatment and Treatment Outcome

In our study, treatment methods for patients were as follows: 123 patients received chemoradiotherapy, 47 patients received chemotherapy, 3 patients received radiotherapy alone, and 3 patients received best supportive care (BSC) alone. Treatment methods for patients were decided in medical conferences between a digestive surgeon, an endoscopist, a clinical oncologist, and a radiotherapy physician. Chemotherapy comprised 5-fluorouracil and cisplatin (FP regimen), docetaxel and 5-fluorouracil and cisplatin (DCF regimen), or combined therapy with 5-fluorouracil and nedaplatin (5FU + NDP regimen). In chemoradiotherapy, only the FP regimen was used as chemotherapy. The radiation dose ranged from 30 to 60 Gy depending on the condition of the patients. Cisplatin was administered at a dose of 70 mg/m^2^ by slow-drip infusion on days 1 and 29. Administration of 5-fluorouracil was performed at a dose of 700 mg/m^2^/day by continuous infusion for 24 h on days 1–4 and days 29–32 in concurrent chemoradiotherapy. Chemotherapy alone was performed with the FP regimen for 34 patients, the DCF regimen for 9 patients, and the 5FU + NDP regimen for 4 patients. In this study, we defined “withdrawn cases” as patients for whom the planned treatment was discontinued for some reason and never resumed and “completed cases” as patients for whom the planned treatment was completed.

Adverse event was assessed according to the Common Terminology Criteria for Adverse Events (CTCAE) Version 4.0. Objective tumor response was assessed by CT scans in accordance with the Response Evaluation Criteria in Solid Tumor (RECIST version 1.1). In this study, we assessed the best overall response, which is the best response recorded from the start of the treatment until the disease progression, and evaluated the disease control rate, which is the sum of complete response plus partial response plus stable disease.

### 2.3. Measurement of Body Composition

Skeletal muscle volume, visceral fat mass, and subcutaneous mass were measured using an enhanced CT image that had been taken for the purpose of evaluating esophageal cancer before treatment. Muscle area at the third lumbar vertebra (L3) is a standard skeletal landmark that correlates with whole-body muscle volume [22]. Cross-sectional skeletal muscle area at L3 was therefore measured from abdominal CT using the Volume Analyzer Synapse VINCENT 3 image analysis system (Fujifilm Medical, Tokyo, Japan). Skeletal muscle was identified and quantified using CT attenuation values with thresholds of −29 Hounsfield units (HU) to 150 HU. To standardize the area according to patient stature, skeletal muscle area at the L3 level (cm^2^) was divided by the square of the height (m^2^), giving the skeletal muscle mass index (SMI) (cm^2^/m^2^) [23,24]. Anonymized CT image were analyzed by two trained study authors (O.S., Y.I.), who were blinded to their outcomes. These authors were trained to use the system to achieve a good level of agreement. We defined sarcopenia based on a SMI cut-off of <52.4 cm^2^/m^2^ for males and <38.5 cm^2^/m^2^ for females [25]. In the same manner, the cross-sectional areas of visceral fat mass and subcutaneous mass at the umbilical level were measured using a built-in function in the Synapse VINCENT system. Visceral fat and subcutaneous fat were quantified within a range of −200 to −50 HU.

### 2.4. Patient Data

Clinical data for all patients were collected from the prospectively maintained database at our instrument. Pathological classification was based on the Guidelines for Diagnosis and Treatment of Carcinoma of the Esophagus 2017 by the Japan Esophageal Society [26]. Height, weight, and body mass index (BMI) were as at the time of first hospitalization. Laboratory data were as at the first visit to our institution. Mortality data were collected via a hospital coding system and by contacting patients’ general practitioner. Mortality dates was determined from the date of first hospitalization until death or the censor date of the study.

### 2.5. Nutritional Screening

We used the subjective global assessment (SGA) proposed by Baker et al. [27] for nutritional screening in this study. SGA has been reported as a method to predict the duration of hospitalization for patients with gastrointestinal cancer [28] and includes components from the medical history and physical examination. Medical history consisted of 4 categories: weight loss, gastrointestinal symptoms, functional capacity, and comorbidities. Physical examination included loss of subcutaneous fat, muscle wasting, and edema. Based on overall SGA questions, patients were classified into 3 groups: A, well nourished; B, mildly to moderately malnourished; or C, severely malnourished. In our hospital, SGA was evaluated by the registered dietitian in all patients on admission.

### 2.6. Statistical Analysis

Statistical analysis was performed using JMP version 9.0.2 software (SAS Institute, Cary, NC, USA). Continuous variables are expressed as median and range, and differences between median values were nonparametrically analyzed using the Mann–Whitney U test. Categorical variables are given as the number of patients, and differences in distributions between groups were tested by the chi-squared test. Survival curves were estimated using the Kaplan–Maier method and compared using the log-rank test. Uni- and multivariate Cox regression analyses were performed to clarify whether sarcopenia predicted OS, and variables showing values of *p* < 0.05 on univariate analyses were subjected to multivariate analysis. Statistical significance was declared for values of *p* < 0.05.

## 3. Results

### 3.1. Patient Characteristics

Characteristics of patients in the sarcopenic and non-sarcopenic groups are shown in Table 1. A total of 101 patients (57.4%) showed sarcopenia before treatment. The prevalence of sarcopenia was significantly higher in males than in females (*p* < 0.01). Brinkman index (number of cigarettes smoked each day × years of smoking) of sarcopenic group was significantly higher than the non-sarcopenic group (*p* < 0.01). The rate of cStage IVb was significantly higher in the sarcopenic group than in the non-sarcopenic group (*p* = 0.01). However, no differences in age, performance status (PS), or alcohol intake were seen between the sarcopenic and non-sarcopenic groups. Likewise, no significant differences were identified in tumor factors, such as tumor location or tumor size. On the other hand, in terms of nutritional factors, BMI, visceral fat mass, and subcutaneous fat mass were all significantly lower in the sarcopenic group (*p* < 0.01 each). Serum albumin was significantly lower, and the SGA indicated malnutrition (SGA B or C) significantly more often in the sarcopenic group (*p* < 0.01 each). The rate of weight loss during the six months preceding treatment was higher in the sarcopenic group than in the non-sarcopenic group (*p* = 0.01).

### 3.2. Treatment-Related Factors

Treatment-related factors for the sarcopenic and non-sarcopenic group are shown in Table 2. No significant differences were seen in treatment methods or regimens in the chemotherapy or disease control rate, or in the incidence of side effects of Grade 3 or more.

Among the withdrawn cases, reasons for treatment withdrawal in the sarcopenic group were aspiration pneumonia in 64.5% (*n* = 20) and significant general condition deterioration in 29.0% (*n* = 9). Reasons in the non-sarcopenic group included significant general condition deterioration in 50% (*n* = 4), aspiration pneumonia, hematemesis, or hemoptysis due to esophagotracheal fistula.

The prevalence of withdrawn cases was significantly higher in the sarcopenic group compared to the non-sarcopenic group. We further examined differences within the sarcopenic group between those patients who could not continue their planned treatment for some reason (withdrawn group; *n* = 30) and patients who completed the planned treatment (completed group; *n* = 69), except for cases receiving BSC (*n* = 3) (Table 3). Serum albumin level and malnutrition according to SGA were significantly lower in A than in B.

### 3.3. Overall Survival Rate

Mean duration of follow-up after treatment was 20.3 months. Eighty-two patients in the sarcopenic group and 63 patients in the non-sarcopenic group died during follow-up. All patients died of esophageal cancer. The OS rate is shown in Figure 1. The survival rate was significantly lower in the sarcopenic group than in the non-sarcopenic group. The 2-year OS rate was 9.8% in the sarcopenic group and 23.7% in the non-sarcopenic group (*p* < 0.01). OS rates in withdrawn cases and completed cases are shown in Figure 2. The OS rate was significantly lower in withdrawn cases than in completed cases. The 2-year survival rate was 0.0% in withdrawn cases and 24.4% in completed cases (*p* < 0.01).

### 3.4. Prognostic Factors for OS

Univariate analysis for OS showed that PS 2–3, cStage IVb, and prevalence of sarcopenia were significantly associated with poor overall survival. Cox proportional hazard regression modeling for OS identified prevalence of sarcopenia as an independent prognostic factor for poor OS in these patients (hazard rate (HR): 1.48, 95% confidence interval (CI): 1.04–2.10) (Table 4).

## 4. Discussion

In this study, we investigated the prevalence of sarcopenia and the impact of sarcopenia on OS in patients with unresectable advanced esophageal cancer. A total of 101 patients (57.4%) with unresectable advanced esophageal cancer were considered to show sarcopenia, and the presence of sarcopenia before treatment was the only independent prognostic factor. To the best of our knowledge, this is the first study to identify pretreatment sarcopenia as showing an independent association with poor prognosis in unresectable advanced esophageal cancer.

Sarcopenia is characterized by loss of skeletal muscle mass, skeletal muscle strength, and physical performance [5]. This phenomenon has been shown to impair physical performance and survival in geriatric, noncancer populations [25,29,30] and to impair survival in a variety of clinical conditions, such as cancer. In esophageal cancer, most studies have evaluated patients before surgery, and sarcopenia has commonly been encountered at a widespread prevalence (from about 16% to 79%) [13]. Sarcopenia has been demonstrated to be significantly correlated with postoperative major complications [21,31]. However, the prognostic value of sarcopenia for patients with surgically treated esophageal cancer has not been clearly established, despite wide investigations using both cohort studies and systematic reviews, although with limited sample sizes and a heterogeneity of approaches to body composition assessment. Recently, two large systematic reviews and meta-analyses of preoperative sarcopenia in patients with esophageal cancer were published [19,20]. The first report examined studies describing the assessment of body composition in 3193 patients with esophageal cancer or gastroesophageal junction cancer from 29 studies. The results showed that sarcopenic patients had a higher incidence of postoperative pulmonary complications (odds ratio (OR): 2.03, 95% CI: 1.32–3.11) after esophagectomy. A meta-analysis of six studies presenting long-term outcomes after esophagectomy identified significantly worse survival in patients who were sarcopenic (HR: 1.70, 95% CI: 1.33–2.17) and concluded that assessment of body composition has the potential to become a clinically useful tool that could support decision-making in patients with esophageal cancer. Another report described an updated comprehensive systematic review and meta-analysis of 1520 patients in 11 cohort studies to investigate the impact of preoperative sarcopenia on survival of patients with surgically treated esophageal cancer. Patients with sarcopenia showed significantly lower 3-year (51.6% vs. 65.4%) and 5-year OS rates (41.2% vs. 52.2%) than those without sarcopenia. Moreover, preoperative sarcopenia was found to be an independent predictor of poor OS (HR: 1.58, 95% CI: 1.35–1.85) and DFS (HR: 1.46, 95% CI: 1.12–1.90) in patients with surgically treated esophageal cancer, regardless of whether patients received preoperative treatment or how sarcopenia was defined. That study concluded that preoperative sarcopenia represents an independent unfavorable prognostic factor for esophageal cancer patients after esophagectomy.

However, little is known about the impact of sarcopenia prior to treatment on the prognosis of patients with unresectable esophageal cancer. Sato et al. [32] reported that the OS rate was significantly lower in sarcopenic patients than in non-sarcopenic patients (3-year OS rates: 36.95% vs. 63.9%). Although univariate analysis for OS showed presence of sarcopenia as significantly associated with poor OS, the Cox proportional hazard regression model for OS did not show sarcopenia as an independent prognostic factor for poor OS in these patients. They therefore concluded that sarcopenia prior to treatment may worsen the long-term survival of patients with unresectable advanced esophageal cancer. However, that study only investigated unresectable, locally advanced esophageal cancer receiving definitive chemoradiation therapy (that is, patients had no distant metastases), and clinical stage was diagnosed as stage IIIc in the TNM classification.

The present study provides a first look into the impact of sarcopenia on patients with unresectable advanced esophageal cancer. Moreover, among patients with sarcopenia, we investigated differences in outcome between patients who could not continue the planned treatment (withdrawn cases) and those who could (completed cases). The OS rate was significantly lower in withdrawn cases than in completed cases (1-year OS: 11.0% vs. 59.9%, respectively). In withdrawn cases, reliable nutritional assessment parameters, such as albumin and SGA, were significantly lower compared to completed cases, and the most common reason for withdrawal was aspiration pneumonia (64.5%). In cases of aspiration pneumonia, deterioration of activity-, disease-, and nutrition-related sarcopenia of generalized skeletal muscles and swallowing muscles may develop into sarcopenic dysphagia [33]. Furthermore, among patients with advanced esophageal cancer, aspiration pneumonia results from not only sarcopenic dysphagia but also malignant obstruction and recurrent nerve paralysis. Multiple factors, such as sarcopenic dysphagia, malnutrition, and disease progression, would contribute to aspiration pneumonia and poor prognosis.

Sarcopenia cannot be sufficiently identified from abnormalities of weight or BMI, and quantification of SMI is necessary to identify sarcopenia. Because CT is often used to evaluate the whole body before treatment, measurement of SMI from this modality is clinically convenient. Evaluating and starting therapy for sarcopenia before cancer treatment can be expected to reduce sarcopenic dysphagia and reduce the frequency of terminated cases. Such interventions can thus also be expected to prevent the progression of sarcopenia that would have resulted from exacerbated dysphagia.

Rehabilitation and nutritional therapy are said to be useful for sarcopenic dysphagia [34]. Implementing other measures for nutritional management of malignant obstruction or recurrent nerve paralysis would also be useful.

In general, treatment of sarcopenia also involves nutritional and rehabilitation therapy. Protein, amino acids, and vitamin D [35,36,37] are considered useful as nutritional therapy. In addition, for cachexia due to cancer, it is possible to use a combination of agents to counteract anorexia and malnutrition, long-term activation of systemic inflammation, and physical inactivity (i.e., progestational agents, nutritional counseling, eicosapentaenoic (EPA)-enriched nutritional supplements, artificial nutrition, L-carnitine, thalidomide, cyclooxygenase-2 (COX-2) inhibitors, and daily exercise). In addition, the use of selective androgen receptor modulators and selective androgen receptor modulators (SARMs) has been reported to be effective. Treatment of sarcopenia may be considered mandatory for esophageal cancer in the near future [38].

Currently, fat mass is recognized as an active endocrine and metabolic organ. Adipocytes produce various mediators, including adipokines, cytokines, and growth factors, which are related to chronic systemic inflammation and can play a role in carcinogenesis, growth, progression, and metastasis [39,40,41,42]. Furthermore, there seem to be functional differences in their distribution. Visceral fat is considered to play a more important role in chronic systemic inflammation and aggressive cancer behavior than subcutaneous fat [43,44]. The clinical impact of fat distribution on prognosis in patients with esophageal cancer is controversial. It was reported that visceral obesity was associated with lymphatic invasiveness and poor response to preoperative treatment in patients with esophageal squamous cell carcinoma [45]. On the other hand, it was reported that low visceral fat content was associated with poor prognosis in patients with upper gastrointestinal cancers, including in esophageal cancer [46]. In this study, visceral fat was not associated with prognosis. However, further analysis is required to clarify the effect of fat distribution on disease progression in patient with esophageal cancer.

Some limitations to this study must be considered. First, this study was a retrospective investigation at a single institution. Second, sarcopenia was defined as a reduction in skeletal muscle mass assessed by CT alone, although EWGSOP (European Working Group on Sarcopenia Older Persons) recommended that muscle strength or walking speed should be considered in addition to them [5]. In this retrospective study, it was impractical to obtain data concerning muscle strength or walking speed from all patients. However, the predominance of CT-based measures relates to their availability in routine practice and high precision and specificity for muscle and fat distribution. Therefore, results of this study will help make generalization among patients with unresectable advanced esophageal cancer. Third, the optical cut-off values for sarcopenia diagnosis remain a matter of debate. In this study, we defined sarcopenia as SMI <52.4 cm^2^/m^2^ for males and <38.5 cm^2^/m^2^ for females, as proposed by Prado et al. [25]. These cut-off values have been widely used to assess the relationship between sarcopenia and surgical outcomes in patients with cancer [13,15,16,18,47]. However, these cut-off values might be unsuitable for Japanese patients with unresectable advanced esophageal cancer. The skeletal muscle mass is well known to be lost in patients with esophageal cancer than in those with cancer of other organs because the former patients are likely to experience weight loss and have a higher risk of malnutrition than the latter [48]. Recently, Asian sarcopenia criteria were determined by AWGS [49] in 2015, and the Japanese sarcopenia criteria were established by the Japanese Hepatic Society in 2016 [50]. However, these criteria were not specialized to patients with cancer as a matter of course. The appropriate cut-off values might be required in each disease category and in each race.

## 5. Conclusions

The presence of sarcopenia was associated with survival outcomes and was an independent prognostic factor in patients with unresectable advanced esophageal cancer. Moreover, terminated cases, for whom planned treatment was discontinued for some reason, frequently had sarcopenia, were malnourished, and showed poor prognosis.

Evaluating sarcopenia before treatment and starting intervention for sarcopenia may improve outcomes for unresectable esophageal cancer.

## Figures and Tables

**Figure 1 jcm-08-01647-f001:**
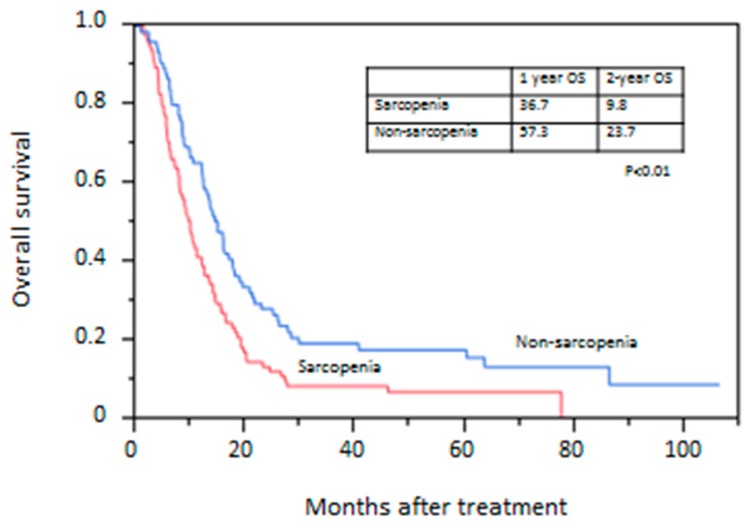
Kaplan–Meier curve of overall survival between patients with and without sarcopenia.

**Figure 2 jcm-08-01647-f002:**
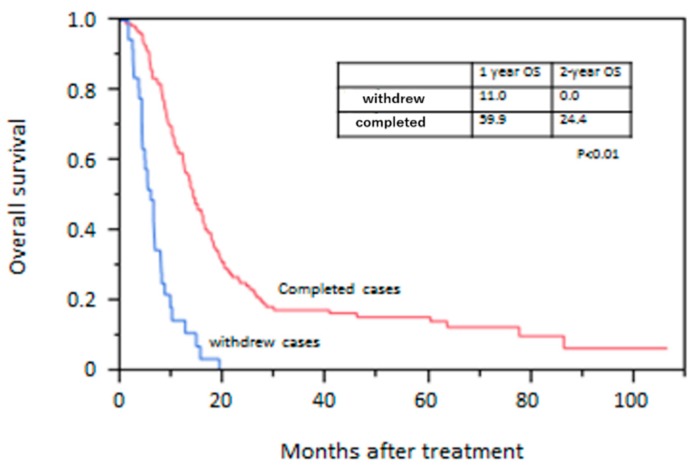
Kaplan–Meier curve of overall survival between withdrawn and completed cases.

**Table 1 jcm-08-01647-t001:** Baseline characteristics of patients.

Variables	Sarcopenia	Non-Sarcopenia	*p*-Value
	(*n* = 101)	(*n* = 75)	
Age (years)	65.1 ± 6.1	65.3 ± 6.2	0.84
Sex (male/female)	96/5	54/21	<0.01
PS (0 or 1/2 or 3)	91/10	73/2	0.08
Brinkman index	952 ± 723	624 ± 482	<0.01
Alcohol drinking, *n* (%)	93 (92.1)	67 (89.3)	0.6
cStage (IVa/IVb)	34/67	40/35	0.01
Primary tumor location (Ce/Ut/Mt/Lt)	7/21/41/32	2/10/45/18	0.06
Tumor length (mm)	60.0 ± 31.8	57.2 ± 28.9	0.54
Body mass index (kg/m^2^)	19.2 ± 2.8	22.3 ± 3.0	<0.01
Visceral fat mass (cm^2^)	54.2 ± 44.9	89.7 ± 60.7	<0.01
Subcutaneous fat mass (cm^2^)	49.1 ± 65.8	84.7 ± 44.8	<0.01
Albumin (g/dL)	3.85 ± 0.46	4.08 ± 0.37	<0.01
SGA (A/B or C)	49/52	63/12	<0.01
Body weight loss rate (%)	6.7 ± 7.7	4.2 ± 5.8	0.01

Values are expressed as mean ± standard deviation. PS, performance status; SGA, subjective global assessment; Ce, cervical esophagus; Ut, upper thoracic esophagus; Mt, middle thoracic esophagus; Lt, lower thoracic esophagus.

**Table 2 jcm-08-01647-t002:** Treatment characteristics and adverse events.

Variables	Sarcopenia	Non-Sarcopenia	
	(*n* = 101)	(*n* = 75)	*p*-Value
Treatment (CT/CRT/RT/BSC)	28/69/2/2	19/54/1/1	0.94
Chemotherapy regimen (FP/DCF/5FU + NDP)	89/6/2	65/4/4	0.48
Disease control rate *, *n* (%)	64 (65.3)	51 (68.9)	0.67
Withdrawn cases, *n* (%)	30 (31.3)	8 (10.9)	<0.01
Aspiration pneumonia	20	1	
General condition deterioration	9	4	
Deterioration of other disease	1	0	
Others	0	3	
Adverse events (≥grade 3), *n* (%)	32 (31.7%)	19 (25.3%)	0.4
Hematological			
Neutropenia	26	18	0.86
Increased creatinine	1	2	0.57
Nonhematological			
Febrile neutropenia	14	5	0.19
Anorexia	6	1	0.09

CT, chemotherapy; CRT, chemoradiation therapy; RT, radiotherapy; BSC, best supportive care; FP, fluorouracil (5FU) + cisplatin; DCF, docetaxel + cisplatin + fluorouracil; NDP, nedaplatin. * assessed by best overall response.

**Table 3 jcm-08-01647-t003:** Patient characteristics according to continuation of treatment.

Variables	Withdrawn	Completed	
	(*n* = 30)	(*n* = 69)	*p*-Value
Age (years)	65.2 ± 6.8	65.1 ± 5.93	0.94
Gender (male/female)	29/1	65/4	1
PS (0 or 1/2 or 3)	2/28	7/62	1
Brinkman index	990 ± 679	932 ± 754	0.72
Alcohol drinking, *n* (%)	27 (90.0)	64 (92.8)	0.64
Body mass index (kg/m^2^)	18.6 ± 3.10	19.5 ± 2.63	0.11
Visceral fat mass (cm^2^)	47.8 ± 42.0	57.0 ± 42.1	0.38
Subcutaneous fat mass (cm^2^)	40.7 ± 51.3	53.2 ± 71.5	0.42
Albumin (g/dL)	3.63 ± 0.39	3.94 ± 0.47	<0.01
SGA (A/B or C)	6/24	42/24	<0.01
Body weight loss rate (%)	8.8 ± 8.4	5.8 ± 7.4	0.07

Values are expressed as mean ± standard deviation. PS, performance status; BMI, body mass index; Alb, albumin; SGA, subjective global assessment.

**Table 4 jcm-08-01647-t004:** Uni- and multivariate Cox regression analyses for overall survival.

Variables			Univariate		Multivariate	
		n	HR (95% CI)	*p*-Value	HR (95% CI)	*p*-Value
Age	<70	131	1			
	≥70	45	1.29 (0.88–1.85)	0.18		
Gender	Female	26	1			
	Male	150	1.48 (0.95–2.43)	0.07		
PS	0–1	164	1		1	
	2–3	12	2.53 (1.19–4.71)	0.01	1.94 (0.89–3.70)	0.08
Brinkman index	>620	107	1			
	≤620	69	1.01 (0.73–1.42)	0.91		
Alcohol drinking	No	16	1			
	Yes	160	1.45 (0.84–2.78)	0.19		
Body mass index (kg/m^2^)	>23	43	1			
	≤23	133	1.42 (0.97–2.11)	0.06		
cStage	IVa	74	1		1	
	IVb	102	1.28 (0.91–1.81)	0.02	1.28 (0.91–1.82)	0.15
Sarcopenia	Absent	75	1		1	
	Present	101	1.63 (1.17–2.29)	<0.01	1.48 (1.04–2.10)	0.03
Visceral fat mass	>55	95	1			
	≤55	81	1.33 (0.94–1.89)	0.09		
Body weight loss rate (%)	<10	136	1			
	≥10	40	1.29 (0.86–1.88)	0.21		

PS, performance status; BMI, body mass index.

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
