# Peer review of "Prognostic Significance of Sarcopenia in Patients with Unresectable Advanced Esophageal Cancer"

_jcm, 2019, doi:10.3390/jcm8101647_

Round 1
Reviewer 1 Report
The article “Prognostic Significance of Sarcopenia in Patients with Unresectable Advanced Esophageal Cancer” by Onishi and collaborators aims to evaluate sarcopenia as a prognostic factor for patients with unresectable advanced esophageal cancer. They recruited 176 patients undergoing some kind of chemo and /or radiotherapy treatment and evaluated treatment response. Body composition was measured by CT image, patient data was collected from the prospectively maintained database and nutritional screening was performed using subjective global assessment (SGA). They observed that sarcopenia was an independent poor prognostic factor for unresectable advanced esophageal cancer patients. The findings are very interesting; however, I have some questions/observations:
Minor changes
Correct the following texts: Line 58: about 50% of patients presenting with.. TO: about 50% of patients with esophageal... Line 103: this study, we defined terminated cases... TO: this study, we defined “terminated cases”... Line 104: discontinued for some reason and never resumed, and continued cases as patients... TO: discontinued for some reason and never resumed, and “continued cases” as patients... Line 148: … p<0.05 on univariate analyses ><0.05 on univariate analyses TO … P<0.05 on univariate analyses
The text in the lines 108-111 is very confusing, please rewrite it: In this study, we assessed the best overall response which is the best response recorded from the start of the treatment until the disease progression, and evaluated the disease control rate which is the sum of complete response plus partial response plus stable disease. The text in the lines 130-131 is very confusing, please rewrite it: Laboratory data were as of the first visit to our institution. Insert at the bottom of the Table I the meaning of Ce/Ut/Mt/Lt. What does “*best overall response” on line 185 mean? Add p-value in Figures 1 and 2 and the symbol of % in the tables of these figures. Remove gridlines from Table 4. Added reference (Cruz-Jentoft et al., 2010) in line 308, after the phrase: “In this study, we used the definition of EWGSOP (European Working Group on Sarcopenia 10 308 Older Persons).” Line 246: The authors cited Sato et al., however, this reference is not included in the Reference session.
Major changes
Question 1
The authors said that sarcopenia was defined by the authors “based on an SMI cut-off <52.4 cm2/m2 for males and < 38.5 cm2/m2 for females” according Prado et al., 2008. However, Prado and colleagues used these cut-off values based on their study population data. The following is a copy of part of the article methodology explaining how those authors calculated the cut-off values for skeletal muscle index (SMI): “Therefore, we used L3 skeletal muscle index to establish our own cut-off s. We used optimum stratification by SAS (version 9.1.3) to find the most significant p value by use of the log-rank χ² statistic to define the sex-specific cut-off s associated with mortality in our patients. Optimum stratification solves the threshold value of the continuous covariate (skeletal muscle index) which, based on log-rank statistics, best separates patients with sarcopenia and those who did not have sarcopenia with respect to time to an event outcome (mortality). These cut-offs were then used to classify patients as those with sarcopenia or those who did not have sarcopenia”. Given the above considerations, the values used as SMI cut-off in this study are not justified, needing explanations.
Question 2
Regarding the definition of sarcopenia, it is not clear why the authors using EWGSOP definition instead of Asian sarcopenia criteria or Japanese sarcopenia criteria. Moreover, EWGSOP suggested an algorithm for sarcopenia case finding in older individuals (Cruz-Jentoft et al., 2010) that the cut-off age is 65 years. Why the authors used 70 years? In addition, there are several criteria for defining sarcopenia according to EWGSOP that were not addressed in this study.
Question 3:
The authors do not discriminate the histology of the tumor and this could have relevance in the development of sarcopenia, because adenocarcinoma and esophageal squamous cell carcinoma have distinct characteristics, such as risk factors, affected esophagus third, affected population. It would be important to discriminate histological types and analyze these data in this study. The study only in patients with esophageal squamous cell carcinoma showed an association between decreased skeletal muscle mass after neoadjuvant therapy with poor prognosis (Motoyama et al., 2016).
Question 4
The authors said in Discussion (lines 223-225) that “However, the prognostic value of sarcopenia for patients with surgically treated esophageal cancer has not been clearly established, despite wide investigations using both cohort studies and systematic reviews, but with limited sample sizes and a heterogeneity of approaches to body-composition assessment.” However, there are several studies, which are cited by the authors, using a larger number of samples than the present study (for example: Elliott et al., 2017).
Question 5
Why did the authors present overall survival rate before the prognostic factor for OS in the results? It is necessary first to know which factors impacted in overall survival (univariate analysis) and which are independent factors (multivariate analysis).
Author Response
Herein we would like to resubmit our manuscript entitled “Prognostic Significance of Sarcopenia in Patients with Unresectable Advanced Esophageal Cancer” to the Journal of Clinical Medicine. We thank the editors and the reviewers for their thorough review of our manuscript. We made revisions as the below letters.
Thank you very much for reviewing our manuscript and offering valuable advice.
We would like to give a point-by-point reply to reviewer’s comments, and revised the manuscript accordingly.
REVIEWERS' COMMENTS TO THE AUTHOR:
Reviewer: 1
Minor changes
Correct the following texts: Line 58: about 50% of patients presenting with.. TO: about 50% of patients with esophageal... Line 103: this study, we defined terminated cases... TO: this study, we defined “terminated cases”... Line 104: discontinued for some reason and never resumed, and continued cases as patients... TO: discontinued for some reason and never resumed, and “continued cases” as patients... Line 148: … p<0.05 on univariate analyses TO … P<0.05 on univariate analyses
Response: According to your suggestion, we revised all of them.
The text in the lines 108-111 is very confusing, please rewrite it: In this study, we assessed the best overall response which is the best response recorded from the start of the treatment until the disease progression, and evaluated the disease control rate which is the sum of complete response plus partial response plus stable disease.
Response: According to your suggestion, we rewrote them.
The text in the lines 130-131 is very confusing, please rewrite it: Laboratory data were as of the first visit to our institution.
Response: According to your suggestion, we rewrote them.
Insert at the bottom of the Table I the meaning of Ce/Ut/Mt/Lt.
Response: According to your suggestion, we added an explanation for abbreviation of Ce/Ut/Mt/Lt at the bottom of Table 1.
What does “*best overall response” on line 185 mean?
Response: We explain the meaning of “*best overall response” in the sentence in the lines 108-111. So, we added * at “Disease control rate*”, and *“assessed by” in the Table2
Add p-value in Figures 1 and 2 and the symbol of % in the tables of these figures.
Response: We added p-value in Figures 1 and 2 and the symbol of % in the tables of these figures.
Remove gridlines from Table 4.
Response: We removed gridlines from Table4.
Added reference (Cruz-Jentoft et al., 2010) in line 308, after the phrase: “In this study, we used the definition of EWGSOP (European Working Group on Sarcopenia 10 308 Older Persons).”
Response: According to your suggestion, we added the reference.
Line 246: The authors cited Sato et al., however, this reference is not included in the Reference session.
Response: It’s our mistake, we added the reference.
Major changes
Question 1
The authors said that sarcopenia was defined by the authors “based on an SMI cut-off <52.4 cm2/m2 for males and < 38.5 cm2/m2 for females” according Prado et al., 2008. However, Prado and colleagues used these cut-off values based on their study population data. The following is a copy of part of the article methodology explaining how those authors calculated the cut-off values for skeletal muscle index (SMI): “Therefore, we used L3 skeletal muscle index to establish our own cut-off s. We used optimum stratification by SAS (version 9.1.3) to find the most significant p value by use of the log-rank χ² statistic to define the sex-specific cut-off s associated with mortality in our patients. Optimum stratification solves the threshold value of the continuous covariate (skeletal muscle index) which, based on log-rank statistics, best separates patients with sarcopenia and those who did not have sarcopenia with respect to time to an event outcome (mortality). These cut-offs were then used to classify patients as those with sarcopenia or those who did not have sarcopenia”. Given the above considerations, the values used as SMI cut-off in this study are not justified, needing explanations.
Response: The cut-off value of sarcopenia using CT has not been settled in the EWGSOP etc. So we used the cut-off values proposed by Prado and colleagues, because those values were widely used to assess the relationship between sarcopenia and surgical outcomes in patients with esophageal cancer such as Sierzega M, et al; J surg Oncol 2019, Elliott JA, et al; Ann Surg 2017, Yip C et al, Eur Radiol 2014, Siegel SR, et al, Am J Surg 2018, Mayanagi S, et al. Ann Surg Oncol 2017, etc..
So, we added sentence in the limitation, line 315-325.
Question 2
Regarding the definition of sarcopenia, it is not clear why the authors using EWGSOP definition instead of Asian sarcopenia criteria or Japanese sarcopenia criteria. Moreover, EWGSOP suggested an algorithm for sarcopenia case finding in older individuals (Cruz-Jentoft et al., 2010) that the cut-off age is 65 years. Why the authors used 70 years? In addition, there are several criteria for defining sarcopenia according to EWGSOP that were not addressed in this study.
Response: As the reviewer mentioned, it is unclear why we used EWGSOP definition. So we added the reason in the limitation, line 308-326. The age of 70 years was a value calculated from the ROC curve of our data.
Question 3:
The authors do not discriminate the histology of the tumor and this could have relevance in the development of sarcopenia, because adenocarcinoma and esophageal squamous cell carcinoma have distinct characteristics, such as risk factors, affected esophagus third, affected population. It would be important to discriminate histological types and analyze these data in this study. The study only in patients with esophageal squamous cell carcinoma showed an association between decreased skeletal muscle mass after neoadjuvant therapy with poor prognosis (Motoyama et al., 2016).
Response: As the reviewer mentioned, it is important to discriminate histological types. In this study, we investigated all esophageal squamous cell carcinoma patients, so we added this to line 84-85 (page2) in addition to line 38 in abstract.
Question 4
The authors said in Discussion (lines 223-225) that “However, the prognostic value of sarcopenia for patients with surgically treated esophageal cancer has not been clearly established, despite wide investigations using both cohort studies and systematic reviews, but with limited sample sizes and a heterogeneity of approaches to body-composition assessment.” However, there are several studies, which are cited by the authors, using a larger number of samples than the present study (for example: Elliott et al., 2017).
Response: We used the term of “limited” sample sizes, compared to two large systematic reviews and meta-analyses of preoperative sarcopenia in patients with esophageal cancer. Of course, in these two large systematic reviews and meta-analyses, Elliott study was including.
Question 5
Why did the authors present overall survival rate before the prognostic factor for OS in the results? It is necessary first to know which factors impacted in overall survival (univariate analysis) and which are independent factors (multivariate analysis).
Response: I’m sorry that I can’t understand what reviewer mentioned. We think that it is common to present the difference in OS between patients with and without sarcopenia before the prognostic factor for OS. In this study, we can find the significant difference between two groups. So we evaluate the prognostic factor effect on OS using Cox regression analyses, and demonstrated that PS 2-3, cStage IVb and prevalence of sarcopenia were significantly associated with poor overall survival in univariate analysis. Finally, we demonstrated that prevalence of sarcopenia as an independent prognostic factor for poor OS in multivariate analysis.

Reviewer 2 Report
The article "Prognostic Significance of Sarcopenia in Patients with Unresectable Advanced Esophageal Cancer” by Onishi and colab. presents the results of a retrospective analysis of all unresectable esofageal cancer patients treated in a Japanese hospital over seven years.
The paper is well written and interesting. However, I have some observations and comments that I believe can help improve the article overall.
Throughout the article, you refer to cases as “terminated”. I do not believe this is routinely used. I suggest replacing throughout the text “terminated” with discontinued/withdrew, progressed or had unacceptable toxicity according to each clinical scenario. Similarly, I do not think the expression “continued cases” should be used – also try replacing it with “patients that received full dose”. I am confused regarding the study design – in the abstract section you state that you reviewed CTs retrospectively, but then in the Material and Method section you state that you recruited patients. A clear study design is very important for a good paper, because it helps you interpret the results and see potential study limits and biases. Please check and clearly state what king of study this isSection 2.3. – measurement of body composition. You should offer some additional details regarding your methodology. Who performed the analysis? Was he/she blind to the outcomes? Did you control for observer variability?
Section 2.5. Nutritional assesment. When was SGA performed? Who performed it?
page 6, row 192-193 – you state that all patients died of esophageal cancer. Are you referring to complications of the esophageal cancer? How did you certify the cause of death for all patients? What were the most frequent causes of death?
Minor comment - page 2, row 93 – please replace “3 patients received palliative treatment” with “3 patients received best supportive care alone” in order to increase clarityAuthor Response
Comments and Suggestions for Authors
The article "Prognostic Significance of Sarcopenia in Patients with Unresectable Advanced Esophageal Cancer” by Onishi and colab. presents the results of a retrospective analysis of all unresectable esophageal cancer patients treated in a Japanese hospital over seven years.
The paper is well written and interesting. However, I have some observations and comments that I believe can help improve the article overall.
Throughout the article, you refer to cases as “terminated”. I do not believe this is routinely used. I suggest replacing throughout the text “terminated” with discontinued/withdrew, progressed or had unacceptable toxicity according to each clinical scenario. Similarly, I do not think the expression “continued cases” should be used – also try replacing it with “patients that received full dose”. I am confused regarding the study design – in the abstract section you state that you reviewed CTs retrospectively, but then in the Material and Method section you state that you recruited patients. A clear study design is very important for a good paper, because it helps you interpret the results and see potential study limits and biases. Please check and clearly state what king of study this is.
Response: Thank you very much for your constructive suggestion. As you mentioned, we replaced “terminated” with “withdrew”, and “continued” with “completed”. I’m sorry that “patients that received full dose” is too long to replace it.
Section 2.3. – measurement of body composition. You should offer some additional details regarding your methodology. Who performed the analysis? Was he/she blind to the outcomes? Did you control for observer variability?
Response: According to your suggestion, we added them in line 134-135.
Section 2.5. Nutritional assesment. When was SGA performed? Who performed it?
Response: According to your suggestion, we added them in line 158-159.
page 6, row 192-193 – you state that all patients died of esophageal cancer. Are you referring to complications of the esophageal cancer? How did you certify the cause of death for all patients? What were the most frequent causes of death?
Response: Mortality data were collected via a hospital coding system and contacting patients’ General practitioner (We added the sentence in line 146-148.). As far as we ingstigated, the main cause of death was esophageal cancer. In withdrew cases, many cases could not continue their planned treatment because of aspiration pneumonia, however they didn’t died because of aspiration pneumonia.
Minor comment - page 2, row 93 – please replace “3 patients received palliative treatment” with “3 patients received best supportive care alone” in order to increase clarity
Response: We replaced as you mentioned.
Submission Date
12 August 2019
Date of this review
23 Sep 2019 11:25:04

Round 2
Reviewer 1 Report
After the corrections,no comments
This manuscript is a resubmission of an earlier submission. The following is a list of the peer review reports and author responses from that submission.
Round 1
Reviewer 1 Report
Onishi et al. investigated the prognostic significance of sarcopenia in patients with unresectable advanced esophageal cancer and found that the pretreatment sarcopenia was an independent prognostic factor. In addition, discontinuation of the planned treatment was more frequent in patients with sarcopenia than in those without. This manuscript is well-written and informative. Only some minor changes are required.
1. I can’t understand the reason why the authors measured visceral fat mass and subcutaneous fat mass. Please mention the purpose in the Materials and Methods section.
2. Was there a difference in the treatment effect between the groups? Please provide the data on treatment effect in Table 2.
3. Page 4, Line 161-162 and Page 5, Line 167-168 say the same thing.